# Rubinstein-Taybi Syndrome: A Model of Epigenetic Disorder

**DOI:** 10.3390/genes12070968

**Published:** 2021-06-24

**Authors:** Julien Van Gils, Frederique Magdinier, Patricia Fergelot, Didier Lacombe

**Affiliations:** 1Reference Center AD SOOR, AnDDI-RARE, INSERM U 1211, Medical Genetics Department, Bordeaux University, Centre Hospitalier Universitaire de Bordeaux, 33076 Bordeaux, France; patricia.fergelot@chu-bordeaux.fr (P.F.); didier.lacombe@chu-bordeaux.fr (D.L.); 2Marseille Medical Genetics, INSERM U 1251, MMG, Aix Marseille University, 13385 Marseille, France; frederique.magdinier@univ-amu.fr

**Keywords:** Rubinstein-Taybi syndrome, *CREBBP*, *EP300*, epigenetics, chromatin, acetylation, chromatinopathies, phenotype, genotype

## Abstract

The Rubinstein-Taybi syndrome (RSTS) is a rare congenital developmental disorder characterized by a typical facial dysmorphism, distal limb abnormalities, intellectual disability, and many additional phenotypical features. It occurs at between 1/100,000 and 1/125,000 births. Two genes are currently known to cause RSTS, *CREBBP* and *EP300,* mutated in around 55% and 8% of clinically diagnosed cases, respectively. To date, 500 pathogenic variants have been reported for the *CREBBP* gene and 118 for *EP300*. These two genes encode paralogs acting as lysine acetyltransferase involved in transcriptional regulation and chromatin remodeling with a key role in neuronal plasticity and cognition. Because of the clinical heterogeneity of this syndrome ranging from the typical clinical diagnosis to features overlapping with other Mendelian disorders of the epigenetic machinery, phenotype/genotype correlations remain difficult to establish. In this context, the deciphering of the patho-physiological process underlying these diseases and the definition of a specific episignature will likely improve the diagnostic efficiency but also open novel therapeutic perspectives. This review summarizes the current clinical and molecular knowledge and highlights the epigenetic regulation of RSTS as a model of chromatinopathy.

## 1. Introduction

Rubinstein-Taybi syndrome (RSTS; OMIM #180849, OMIM #613684), formerly called thumb syndrome and hallux larges, is a rare neurodevelopmental genetic abnormality whose incidence is currently estimated between 1/100,000 and 1/125,000 births [1]. The transmission is autosomal dominant and the vast majority of cases (~99%) occur sporadically *de novo* although a few familial cases have been reported [2,3,4].

This syndrome is now well-defined phenotypically and is characterized primarily by post-natal growth retardation, characteristic facial dysmorphia, large thumbs and hallux, and intellectual deficit [5,6]. There are no pathognomonic criteria for RSTS but there is a broad phenotypic spectrum associated with these cardinal signs. Multiple malformations are reported, including cardiac, genitourinary, digestive, Ear-Nose-Throat (ENT), and skin malformations. Patients also present an increased risk of developing benign tumors [1,5,7,8,9,10,11].

Pathogenic variants in two highly evolutionarily conserved genes have been implicated in the etiology of RSTS: the *CREBBP* gene encoding the cAMP response element-binding protein (CREB) binding protein (NM_600140) located in 16p13.3 [12] and the *EP300* gene encoding the EA1-associated protein p300 (NM_602700) located in 22q13 [13]. These two genes are ubiquitously expressed and encode acetyltransferases with a major role in histone acetylation and chromatin remodeling involved notably in neuronal plasticity and cognition [13,14]. The Rubinstein-Taybi syndrome is a developmental disorder whose physiopathology is based primarily on an epigenetic mechanism, belonging thereby to the group of “Chromatinopathies” defined as Mendelian disorders of the epigenetic machinery, as reviewed in [15].

## 2. Clinical Description

In 1957, the first description of this syndrome was reported by Michail et al. [16], as a case presenting wide thumbs with a radial deviation. However, this syndrome remained relatively unknown until 1960 when J. H. Rubinstein, a pediatrician, and H. Taybi, a radiologist, reported seven children with large thumbs and hallux and minor facial feature and intellectual disability [6]. Since then, this syndrome has been clearly identified as a severe abnormality of embryonic development.

### 2.1. Antenatal Anomalies and Pregnancy

The diagnosis of RSTS is hardly ever made, and very rarely mentioned during pregnancy as there are only a few antenatal signs. Moderate intrauterine growth retardation (IUGR) may be noted as well as polyhydramnios. However, a higher incidence of pre-eclampsia and hypertension in pregnancy of children carrying a pathogenic variant in *EP300* are reported (23% to 33% of cases against 5% to 8% of cases in the general population) [17,18,19,20,21,22,23,24]. The contribution of three-dimensional ultrasonography can improve the detection of typical facial features, but abnormal extremities seem to remain the main diagnostic criteria [24,25,26,27,28,29]. Moreover, brain anomalies, especially cerebellar hypoplasia, and abnormalities of the gallbladder (in 22% of cases) appear to be suggestive antenatal markers [24].

The diagnosis is most often made at birth or in early childhood by observing the classic association of post-natal growth retardation, characteristic facial dysmorphism, broad thumbs and halluces, and intellectual disability.

### 2.2. Facial Dysmorphism

The classical facial appearance in children associates microcephaly, bitemporal retraction, downslanted palpebral fissures, epicanthic folds, arched eyebrows with long eyelashes, ptosis of the eyelids, strabismus, high arched palate, and low set and posteriorly rotated ears. The most characteristic dysmorphic criterion is the pronounced appearance of the nose, which has a broad root, with a long protruding septum and a long columella below the alae nasi. Another evocative criterion is the very characteristic smiling aspect with the closure of the palpebral fissures called “grimacing smile”, especially in the newborn.

This facial dysmorphism only becomes characteristic late in childhood. The facial phenotype is evolutionary, and the appearance is different in the newborn, with more often upslanting palpebral fissures, depression of the root of the nose with hypertelorism, and microretrognathia. A capillary hemangioma is also often described. The typical facial aspect is often obvious in adults. Less frequently, the facial phenotype may include a wide anterior fontanel or delayed closure, frontal bumps, low implantation hairline, deviation of the nasal septum, thin upper lip, small mouth, thin upper helix, or pits in the posterior part of the helix [5,7,9,17,21,30,31,32,33,34,35,36] (Figure 1A).

### 2.3. Distal Limb and Skeletal Abnormalities

Abnormalities of the extremities are usually one of the most characteristic phenotypic elements to evoke the diagnosis (Figure 1B).

Hands are described as short and broad with a characteristic massive thumb that may be spatulated, short and stocky, flat and broad, or simply broad [5]. This thumb abnormality is not constant but is found in between 69% to 97% of cases depending on the studies and the gene involved. The radial deviation of the last phalanx of the thumb is also very suggestive but found in a very heterogeneous way (2 to 88% of cases) [5,7,10,17,30,37,38]. Other hand anomalies include, with a decreasing frequency, large distal phalanges of the other fingers, clinodactyly of the fifth fingers, persistent palmar pads of the distal phalanges, a single transverse palmar crease uni or bilateral, camptodactyly, and ulnar deviation of the second or third fingers [7].

Feet features include an almost constant (between 87 and 100% of cases) very wide hallux, in 11% of cases a duplication of the hallux can be seen on the radiographs, and angulation of the last phalanx in varus (7%) or in valgus (17%) is described [5,7,17,30,37]. The other anomalies observed are flat feet, overlapping toes in half of the cases, widening between the first two toes, and cutaneous syndactyly II-III of the toes. More rarely, clubfoot, post-axial polydactyly, or agenesis of the distal phalanx of the hallux have also been reported [5,7,10,17,30,37,38,39,40,41].

Numerous other skeletal anomalies have been described such as pectus excavatum, costal agenesis, or cervical vertebral anomalies (C1-C2 instability, cervical vertebral fusion) [42]. Hypotonia is frequent. Children may present with congenital or acquired scoliosis, lordosis, or kyphosis [5,7,9,17,21].

### 2.4. Development and Behavior

Intellectual disability in patients with RSTS is almost constant but highly variable with intelligence quotient (IQ) ranging from 25 to 79 [7,9]. Language delay is present in 90% of cases [9]. A few individuals have no verbal language and use sign language or other non-verbal language methods. An interesting and peculiar neurological aspect of RSTS patients is that fluid reasoning is higher than the IQ showing a more flexible cognitive ability in these individuals [43]. Patients with an *EP300* mutation have an overall milder intellectual disability or even normal intellectual efficiency compared to patients with a *CREBBP* mutation [17,44]. The acquisition of walking is delayed, usually around the age of 2 to 3 years, due to constant hypotonia initially.

Behavioral symptomatology includes hyperactivity, noise intolerance, attention and motor difficulties, idiosyncrasies, and maladaptive and unusual behaviors (primarily self-injury) [9,45,46,47]. In addition, specific behaviors are frequently found combining attentional difficulties, motor stereotypies, visio-spatial clumsiness, and visio-motor coordination difficulties [48,49]. Children with RSTS are often described by their families as having sympathetic and cheerful behavior. The behavioral phenotype is age-dependent and changes during adolescence and into adulthood with the emergence of anxiety, obsessive-compulsive disorder, mood instability, autism spectrum disorder, and auto and heteroaggressive behavior [50,51]. Taupiac et al. were able to define a specific developmental profile in which expressive language emerged as a particularly impaired social-emotional ability and was very strongly correlated with many other cognitive and social-emotional functions that had a higher level of development [52].

### 2.5. Growth Retardation and Microcephaly

Children with RSTS most often progress with moderate growth retardation and microcephaly. Intrauterine growth and birth measurements (weight, height, and occipital frontal circumference (OFC)) appear classically around the 50th percentile. The average weight, height, and OFC at birth are respectively 3.300 kg, 49.7 cm, and 34.2 cm for boys and 2.970 kg, 48.6 cm, and 32.2 cm for girls. A delay in bone age is often associated (74%) [31,34]. Microcephaly is a classic feature and is present in 35 to 94% of cases, depending on the study [5,7,17,21,33,34]. There is also a risk of overweight or obesity appearing in childhood in boys and at puberty in girls (Figure 1C). This risk appears to be higher in women since the average adult weight is 61.43 ± 14.89 kg with an average BMI of 26.64 ± 5.5 kg/m^2^ compared to an average final weight in boys of 60.67 ± 13.63 kg with an average BMI of 21.90 ± 3.45 kg/m^2^ [31]. Based on these observations as well as the specificities of RSTS, new specific growth curves were edited in 2014 for height, weight, OFC but also for body mass index (BMI) [31].

### 2.6. Additional Features

Less frequently, many other organ anomalies and malformations have been associated with the syndrome (Figure 1C).

Nonspecific electroencephalogram (EEG) abnormalities are seen in 66–76% of cases, but epileptic manifestations are very rare in patients with RSTS, ranging from almost zero to 25% of cases depending on the study [5,7,53,54]. The most common features found in brain magnetic resonance imaging (MRI) are dysmorphic aspects of the corpus callosum (73.6%). Periventricular posterior white matter abnormalities (63%), dysmorphic aspects of the cerebellar vermis (58%), and olfactory bulb hypoplasia or aplasia (32% of cases) were also observed [43,53,55,56]. More rarely, Arnold Chiari malformation, pituitary hypoplasia, or Dandy-Walker malformations have been reported [5,7,9,41,57,58,59]. Spinal cord malformations have also been described (tethered spinal cord, lipoma, and spina bifida) [7,60].

Of patients with RSTS, 24% to 58% have cardiac anomalies [5,7,9,17,21,37,61,62]. These congenital heart defects range from simple defects (atrial septal defect, ventricular septal defect, patent ductus arteriosus, coarctation of the aorta, aortic bicuspidism, tricuspid atresia, and pulmonary atresia) with or without conduction abnormalities to complex defects including pseudotruncus, left heart hypoplasia, dextrocardia, and single ventricle. In terms of ENT, conductive and/or sensorineural hearing loss may occur in approximately 24% of cases. Recurrent acute otitis media occurs in 50% of cases and is more severe (with risk of perforation) than in the general population [9].

Dental abnormalities are more commonly reported, affecting between 67% and 85.7% of children with RSTS [7,37,38]. There is a significant rate of dental caries in these patients (15–36%). Hypodontia (30%), supernumerary teeth (15%), and persistent milky teeth are described. The most common abnormality found is the presence of talon cusps in 50–70% of RSTS cases compared to approximately 2.5% in the general population representing a diagnostic tool for the clinician [63].

Various ocular features (found in 65–80% of cases) are noticed, the most described being strabismus and the associated risk of amblyopia (60–71%) and refractive anomalies (41–56%). Congenital lacrimal anomalies in RSTS patients range from 10 to 37%. The risk of glaucoma requires early ophthalmologic evaluation in the neonatal period. Other abnormalities include ptosis (29–32%), uni or bilateral colobomas of iris, retina or optic nerve (9–11%), Duane syndrome (8%), and cataract [64,65,66,67].

Feeding problems (71–88%) as well as gastroesophageal reflux disease and constipation (40–74%) are common in young children [5,7,37,38,41,68]. More rarely, cases of megacolon/Hirschsprung’s disease have been reported [57,69]. Cryptorchidism is described in the vast majority of boys (78–100%) [7,9]. In girls, genital anomalies are only sporadic cases [70,71,72].

According to the literature, 24–66% of patients develop renal or urinary tract anomalies. These include renal agenesis, renal or pyloric duplication, nephrotic syndrome, hydronephrosis, or vesicoureteral reflux [5,8,9,10,17,21,37].

Keloid or hypertrophic scars have been described in approximately 10–24% of cases [17,73]. Other dermatological findings include supernumerary nipples in 15% of cases, ingrown toenails or paronychia, and also hypertrichosis (75%) or glabellar hemangioma suggestive in the first weeks of life [8].

Manifestations of immune dysfunctions, affecting mostly B cells are more frequent than in the general population. Saettini et al. have reported on, a cohort of 97 RSTS patients, 72.1% of recurrent or severe infections, 12.3% of autoimmune/autoinflammatory complications, and 8.2% of lymphoproliferation. Syndromic immunodeficiency was diagnosed in 46.4% of patients [74].

To date, a total of 132 tumors have been reported in 115 individuals with RSTS [11]. These are primarily neural crest derived tumors (neuroblastoma, medulloblastoma, oligodendroglioma, meningioma, pheochromocytoma, rhabdomyosarcoma, leiomyosarcoma, seminoma, odontoma, choristoma, hepatoblastoma, and pilomatricoma). Cases of leukemia and non-Hodgkin’s lymphoma have been reported [75,76,77,78,79,80]. The incidence of malignancy in RSTS patients was initially estimated to be between 3% and 10% [10]. A recent study of the Dutch RSTS population found an increased risk to develop meningiomas (8.3%) and pilomatricomas (17.6%) but an increased risk for malignant tumors could not be proven without a clear genotype–phenotype correlation [11].

**Figure 1 genes-12-00968-f001:**
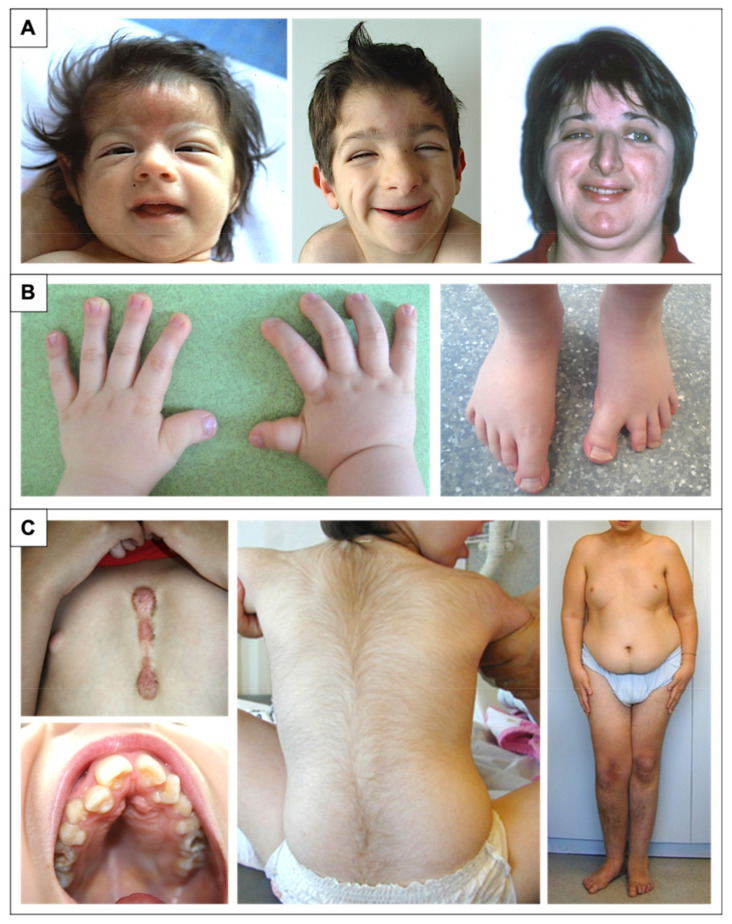
Physical features in RSTS patients. (**A**) Evolution of the phenotype from birth to adulthood. The glabellar hemangioma classically disappears during childhood. The palpebral slits are more oriented downward and outward. Nasal features are more obvious with a prominent nose and a protruding columella. The characteristic grimacing smile with closure of the palpebral fissures and bilateral and asymmetric ptosis of the eyelids can also be noted. The patient on the left was previously reported at 2 months of age by Lacombe et al. [81]. The patients in the middle and on the right have not been reported in any publication to date and were 4 and 33 years old, respectively, at the time of description. (**B**) Distal limb abnormalities with broad thumbs and halluces. Characteristic aspect of short, broad hands with broad thumbs with radial deviation and spatulate last phalanges; enlarged halluces are a near-constant sign. (**C**) Additional classic features in RSTS. We can note the formation of a keloid scar post-sternotomy for cardiac surgery; the highly arched palate with the presence of talon cusps of the four upper incisors and the dental caries of the premolars; the hypertrichosis and the risk of being overweight or obese during adolescence.

## 3. Genotype and Mutation Spectrum

Rubinstein-Taybi syndrome is inherited as an autosomal dominant trait. However, the occurrence is sporadic in the large majority of cases (~99%), with mutation occurring *de novo* in the family. In most families, the index case is the only member with the disease. However, cases of moderately affected relatives by somatic mosaicism have been reported [2,82,83,84] up to familial forms transmitted by one affected parent [2,4,85,86,87], confirming the clinical heterogeneity of the syndrome.

Historically, the location of the first gene involved in RSTS at 16p13.3 was identified by Imaizumi et al. in 1991 [88,89] and confirmed in 1992 by the works of Lacombe et al. [81] and Tommerup et al. [90]. Then, in 1995, Petrij et al. [12] identified this gene as *CREBBP* which encodes the cAMP response element-binding protein (CREB) binding protein. Initially, this protein was given this name because it was described as a partner of the CREB transcription factor [91]. Ten years later, mutations were identified in a *CREBBP* paralog gene, *EP300* as an alternative cause of RSTS [13]. *EP300* encodes the p300 protein that was originally described as a factor interacting with the EA1 protein of adenovirus type 5 [92,93].

The syndrome has been subdivided into type 1 associated with the *CREBBP* mutation spectrum (RSTS1; OMIM #180849) and type 2 associated with the *EP300* mutation spectrum (RSTS2; OMIM #613684). The frequency of abnormalities in the responsible *CREBBP* gene is approximately 55–75% of cases [2,12,13,37,65,94,95,96,97,98,99]. To date, 500 pathogenic variants in this gene have been referenced as causing RSTS1 (55 of which are unpublished) based on the HGMDPro variant and LOVD databases (analyzed on 27 April 2021) [100,101] (Appendix A). The mutational spectrum includes 80.2% point mutations of which 55.2% are truncating mutations, 9.2% splicing mutations, and 16.8% missense mutations, 18.8% correspond to large rearrangements [100,101] (Figure 2A). There are no real hot spot mutations in *CREBBP* with a mutational spectrum distributed along the 31 exons. However, some recurrent mutations have been described and it is noted that about 52% of the reported missense mutations are located in the lysine acetyltranferase (KAT domain) [99]. An exception to this is the presence of an unstable region of *CREBBP* located between introns 1 and 2, characterized by a high frequency of repeated or palindromic sequences resulting in recurrent rearrangements in this region [37,102,103,104]. The presence of these heterozygous mutations or microrearrangements suggests a haploinsufficiency mechanism leading to the developmental abnormalities observed in the syndrome.

Abnormalities in the *EP300* gene are responsible for about 8–11% of cases [2,13,17,18,19,20,21,22,44,99,105,106,107]. To date, 118 pathogenic variants in this gene have been referenced as causing RSTS2 (eight of which are unpublished) based on HGMDPro variant and LOVD databases (on 27 April 2021) [100,101] (Appendix A). The mutational spectrum includes 84.7% point mutations of which 69.5% are truncating mutations, 5.1% splicing mutations, and 13.6% missense mutations for 11.8% large rearrangements [100,101] (Figure 2B). Like *CREBBP*, there is no hot spot mutation in *EP300*, with only four pathogenic variants referenced more than twice in the databases: three in the catalytic domain and one in exon 2 [100,101]. In contrast, almost all of the predicted pathogenic missense mutations of *EP300*-associated RSTS are located in the KAT domain. Only three patients with RSTS have been reported in the literature with a missense mutation in *EP300* out of KAT domain. However, each of these mutations was inherited from a healthy parent, making the pathogenic involvement of these variants difficult.

*CREBBP* and *EP300* contain 31 exons and span approximately 155 kilobases (kb) and 87 kb, respectively [18,107,108]. The CBP (or KAT3A) and p300 (or KAT3B) proteins are paralogous transcriptional coactivators with intrinsic KAT activity. They are the only two members of the KAT3 family due to their low sequence homology to other acetyltransferases in the human genome [109] (Figure 3). They share a similar structure with different functional domains, including different types of protein–protein interaction motifs, and have high sequence identity (58% overall). On the N-terminal side, there is a nuclear receptor interaction domain (NRID or RID), that can bind to PXXP motifs. There are three cysteine-histidine-rich regions (C/H1 to C/H3) involved in protein–protein interactions. The C/H1 and C/H3 domains contain zinc finger transcriptional adapters (TAZ1 and TAZ2), and the C/H3 domain also contains a ZZ zinc finger domain and interacts with the EA1 oncoprotein. C/H2 contains a homeodomain (PHD). The catalytic domain has been highly conserved during evolution with 86% of similarity between CBP and p300. It includes a KAT domain and flanking regions (Bromodomain, C/H2, and C/H3 regions) [110,111]. Other non-catalytic domains show a high sequence similarity. The KIX domain allows the interaction of CREB, specifically at phosphorylated residue 133 (Ser133) of the CREB protein, with other transcription factors, and finally, a bromodomain (BD) links acetylated lysines [112]. On the C-terminal side of p300/CBP, there is an interferon-binding transactivation domain (IBiD), which contains a nuclear binding coactivator domain (NCBD) and a glutamine-rich domain, followed by a proline-containing PxP motif [113,114] (Figure 3).

These two proteins are able to interact with the basic transcription factors, TATA-binding protein (TBP) and TFIIB, and can form a complex with RNApolII [114,115,116]. These interactions allow KAT3 enzymes to play a crucial role in transcription initiation. They are also cofactors for oncoproteins, for viral protein processing (e.g., E1A), or for tumor suppressor proteins (e.g., p53 or BRCA1) [117]. Thus, CBP and p300 promote transcription in two ways: on one hand, they act as a bridge, linking transcription factors binding DNA to the transcription machinery, and on the other hand, by acetylation of histones, they create a chromatin environment favoring gene expression (Figure 4). This histone acetylation will play a key role in transcriptional activation by two distinct molecular mechanisms. Firstly, acetylation will partially neutralize the positive charge of histones, which will weaken their interaction with DNA. This will allow the opening of the chromatin structure and thus facilitate the access of transcription factors to their recognition elements [118]. Secondly, acetylated lysines can create a specific signal for regulatory factors or chromatin remodeling complexes to target a specific region. In contrast, histone deacetylation mediated by histone deacetylase (HDAC) will be associated with the repression of gene expression [119]. Apart from transcription, p300 and CBP act indirectly in other nuclear processes through their interaction with several proteins (which they often acetylate) involved in DNA replication and repair [120,121]. They have also been implicated in the regulation of cell cycle progression through interactions with cyclin E and cyclin-dependent kinase 260, as well as in intranuclear transport through acetylation of importin-α [122].

## 4. Phenotype-Genotype Correlations

Early studies of phenotype-genotype correlations, prior to the discovery of the involvement of the *EP300* gene, initially compared *CREBBP*-mutated patients to patients with a clinical diagnosis without mutations identified in *CREBBP* [38,96,123,124]. The comparison of groups with and without identified mutations, across all these studies, did not reveal significant differences in phenotypic aspects, level of psychomotor or intellectual development, prevalence of organ malformations, or tumor predisposition.

Subsequently, the description of patients with *EP300* mutation allowed comparisons between RTS1-related *CREBBP* and RSTS2-related *EP300* patients. These studies have shown that the phenotype of *EP300* mutation carriers is similar to those of *CREBBP* but in a milder form [18,19,20,22,105,106,107,125]. More recent studies on larger cohorts have confirmed these results by refining the phenotypic aspects. The facial appearance is less marked except for the protruding columella which appears equally frequent in both populations. Extremity anomalies are similar to those seen in *CREBBP*-mutated patients but are less frequent. The exception to this is the absence of radial deviation of the thumb in almost all *EP300*-mutated patients [17,21,44]. Similarly, all degrees of intellectual deficits are observed but in general, the cognitive level is higher in *EP300*-mutated patients. Notably, microcephaly is significantly more observed in *EP300*-mutated patients (83–86% of cases) compared to *CREBBP*-mutated patients (54% of cases). There was no significant difference in the existence of organ malformations between the two groups of patients [17,21,44]. Several studies have reported a higher rate of intra uterine growth retardation (IUGR) as well as cases of pre-eclampsia and gestational hypertension in pregnancies of children carrying a mutation in *EP300* [18,19,20]. The works of Fergelot et al. and Cohen et al. tends to confirm this observation, since 42% to 50% of patients with *EP300* mutations developed IUGR, compared to 25% of the *CREBBP* cohort [17,21]. Furthermore, in the general population, the rate of preeclampsia during pregnancy is estimated to be between 5% and 8% [23] compared with 23% to 33% of *EP300* cases and 3% of *CREBBP* cases. The main clinical findings and their frequency in RSTS1 individuals compared to RSTS2 individuals are summarized in Table 1.

A correlation between the severity of the phenotype and the presence of large exonic deletions or alterations in function-relevant protein domains or leading to a truncated protein before the KAT domain has been initially suggested [38,123,127]. Thanks to recent studies, no significant phenotype/genotype correlation could be shown between the phenotype and the mutation type and location or deletion size for either *CREBBP* or *EP300* genes in RSTS patients [17,37,99,104].

However, a new clinical entity has recently emerged with the identification of missense mutations between the end of exon 30 and the beginning of exon 31 of *CREBBP* and *EP300* related non-RSTS phenotype, referred to as Menke-Hennekam syndrome (MKHK, OMIM #618332) [128,129,130]. Patients present a different syndrome which is not RSTS with severe developmental delay, microcephaly, telecanthus, short upturned palpebral fissures, ptosis, depressed nasal bridge, short nose, short columella, anteverted nares, long deep philtrum, low-set ears with a protruding upper part, and fibular deviation of the distal phalanx. These missense variants are located at the ZNF2 (zinc finger, ZZ type) and ZNF3 (zinc finger, TAZ type) domains, which contain cysteine residues important for Zn2+ binding. These domains are involved in stabilizing a helical fold that provides binding interactions with many transcriptional regulatory proteins [131,132]. These data suggest that this group of mutations specifically affects the binding properties of the two zinc finger domains to different *CREBBP* partners by affecting their own folding.

## 5. Epigenetic Regulation and Cognitive Function in RSTS

Epigenetic mechanisms include a wide range of dynamic processes known to control nuclear functions (transcription, replication, repair, splicing) by orchestrating, at least in part, DNA access to regulatory complexes in a tissue-specific manner. The underlying molecular mechanisms include post-translational modifications of histones, covalent modifications of the DNA itself, and multiple “chromatin remodeling” activities, forming specific combinations of epigenetic marks that define the functional regions of the genome (active/silent promoter, enhancer, barrier elements, etc.) [133]. Histone acetylation is one of the key mechanisms for the global and local control of chromatin structure. In order to understand the phenotypic specificities and elucidate mechanistic insights on cognitive impairment of RSTS patients, different strains of CBP and p300-deficient mice were generated and summarized in this review [14]. More recently, Alari et al. have established iPSC-derived neurons (i-neurons) from peripheral blood samples of three *CREBBP*- and two *EP300*-mutated patients compared to four unaffected controls [134,135].

### 5.1. Syndromic Manifestations in the Mouse Model

Heterozygous mice (*Cbp*^+/−^ and *p300*^+/−^) reproduce some craniofacial aspects associated with RSTS (prominent forehead, blunt nose, and wide anterior fontanel), but large thumbs and hallux, diagnostic criteria for RSTS, are apparently not reproduced in mice [136,137].

In addition to skeletal abnormalities, heterozygous mutants also exhibit growth retardation [136,137,138,139], increased insulin sensitivity [140], cardiac malformations [136], and disorders of hematopoiesis with a tendency to develop hematologic malignancies [137], consistent with the tumor suppressor role of the KAT3 proteins [141] as well as the increased risk of cancer observed in RSTS patients [11].

### 5.2. Histone Acetylation Modifications and Memory Development

In vertebrates and invertebrates, histone acetylation modifications in the neuron nucleus have been associated with memory acquisition. Experiments in mouse models have shown that several nucleosome positions are acetylated during learning tasks. These marks are associated with specific stages of memory development and have been detected in the chromatin of neuronal cells. For example, H3 acetylation increases rapidly in the hippocampus and lateral amygdala after contextual fear conditioning [142,143], while spatial memory acquisition (in a water maze) is associated with H2B and H4 acetylation in the hippocampus [144]. On the other hand, some rodent studies have shown that acetylation changes may be restricted to the promoters of genes known to be involved in memory formation, such as that encoding the neurotrophin BDNF (involved in synaptic plasticity and long-term memory maintenance). For example, histone H3 is hyperacetylated at the promoter of the BDNF gene during the retrieval of conditioned fear memory, this hyperacetylation being necessary for memory reconsolidation [145]; whereas the extinction of this conditioned fear memory is associated with hyperacetylation of H4 in the same promoter in neurons of the prefrontal cortex [146]. These data indicate that histones may be differentially acetylated during memory formation.

Histone acetylation is mediated by lysine KAT activity and is generally associated with gene activation, whereas histone deacetylation, mediated by HDAC, has been associated with repression of gene activity [119]. Thus, KAT and HDAC proteins play a key role in memory development. These proteins are ubiquitously expressed, but some are particularly expressed in brain areas involved in learning and memory, such as the hippocampus or the prefrontal cortex [147]. This is particularly the case for proteins of the KAT3 family (CBP and p300) and KAT2B (also named p300/CBP-associated factor, PCAF) [148,149].

### 5.3. Role of KAT3 Proteins in Neurodevelopment and Cognitive Impairment

CBP and p300 play a crucial role during development by controlling the proliferation and differentiation of different cell lines, including those of the nervous system [150]. In mammals, both proteins are initially required for neural tube closure at the embryonic stage. Homozygous mice *Cbp*^−/−^ or *p300*^−/−^ null alleles die early during embryonic development. Death can occur by exencephaly or by vascular or cardiac malformation. In addition, heterozygous *Cbp*^+/−^ and *p300*^+/−^ double mutations are also lethal [136,139,151]. Notably, loss of function of one of the two proteins in heterozygous mice is compensated by overexpression of the other in later stages of development [138,152,153]. This indicates that these two proteins play redundant roles during early development and that some level of CPB and p300 activity is essential during embryonic development.

Different mouse models [14] were also examined for numerous learning and memory exercises. These experiments revealed impairments in associative memory (conditioned fear or avoidance tasks), episodic memory (object recognition task), and spatial memory (Morris water maze task) as well as in other behavioral aspects related to intellectual disability [154]. Overall, studies have highlighted the importance of CBP and p300 functions in processes related to neural plasticity, including learning and memory [112]. In particular, experiments have been conducted in transgenic mice with regulatable expression of a dominant-negative variant of *CREBBP* with a mutated KAT domain (*Cbp* {HAT^−^} mice). The results suggested that intact KAT activity in the adult prosencephalon was required for the consolidation of some forms of memory [155].

Most CBP mutant strains showed specific deficits in long-term memory but no impact on short-term memory, suggesting a role of the CBP protein in memory consolidation. However, in 2010 Chen et al. studied conditional knock-out (cKO) mice in which CBP was completely inactivated in excitatory neurons in the postnatal forebrain and showed deficits in both long-term and short-term memory [152]. This suggests a role for CBP in both memory storage but also in memory encoding. However, these results are tempered by the fact that this effect on short-term memory has not been replicated in subsequent studies using the same cKO mouse model [153,156]. In addition to memory deficits, several strains with a deficient CBP protein, including *Cbp*^+/−^ heterozygous mice and *Cbp*^KIX/KIX^ homozygous knock-in (KI) mice, carrying a point mutation in the KIX domain blocking the CBP-CREB interaction, had motor coordination disorders [157,158]. This may be similar to the locomotor and coordination impairments observed in patients with RSTS.

In the cellular model, RSTS neurons are able to generate action potentials but are hypoexcitable with an altered morphology showing reduced average branch length and increased branch number. An interesting fact is that the i-neurons carrying a missense mutation causing a defective KAT activity did not show any morphological alteration but intrinsic hypoexcitable features suggesting that mutant CBP protein causes impairment at a later stage of differentiation [134]. Transcriptomic analysis on these RSTS i-neurons revealed two levels of dysregulation during neuronal differentiation. The first concerns “active” gene modulation associated with aberrant upregulation of genes involved in neural migration and axonal and dendritic targeting and downregulation of RNA and DNA metabolic genes. The second relates to a “passive” gene modulation by deregulation of some genes involved in synaptic integration compared to controls [135].

All these data could reveal neuronal biomarkers allowing a better understanding of the cognitive deficit and behavioral disorder of RSTS patients.

### 5.4. RSTS and Related Chromatinopathies

Among rare diseases, genetic neurodevelopmental disorders represent a real diagnostic and therapeutic challenge. More than 10% of them are directly linked to the epigenetic machinery. Thanks to the advent of high-throughput sequencing techniques, significant progress has been made in the discovery of Mendelian disorders and genes related to the epigenetic machinery. To date, 82 human conditions resulting from mutations in 70 genes encoding chromatin-modifying enzymes have been identified and listed [15]. These genes encode “writers” (responsible for establishing a mark on either DNA or histone tail), “erasers” (removal or modification of written marks), “readers” (mediate the interaction of the mark with a protein complex to enhance transcription), or “remodelers” (alter the accessibility of chromatin through ATP-dependent sliding of nucleosomes along DNA). Such disorders, affecting the structure of chromatin genome-wide, are called “chromatinopathies” and show broad phenotypic overlaps suggesting that common networks may be affected.

As previously discussed, histone acetylation is one of the key mechanisms of the epigenetic machinery and RSTS represents a model of a neurodevelopmental disorder with an epigenetic origin. RSTS share clinical overlap with other chromatinopathies, particularly with other disorders caused by aberrant histone acetylation such as Floating Harbor syndrome (FLHS, OMIM #136140) (mutations in the SRCAP gene encoding an SNF2-related chromatin remodeler, coactivator of CBP-mediated transcription), Genitopatellar (GTPTS, OMIM #606170) and Bieseker-Young-Simpson (SBBYSS #603736) syndromes (mutations in *KAT6B* gene encoding the Lysine acetyltransferase 6B, part of the H3 acetyltransferase complex). Gabrielle-De Vries syndrome (GADEVS, OMIM #617557) (mutations of *YY1* gene) has been also associated. GADEVS is not per se a chromatinopathy but YY1 acts indirectly on epigenetic machinery as a key partner encoding transcriptional regulator interacting with CBP/p300) [159]. More recently, Negri et al. and Di Fede et al. have reported 10 cases with an initial clinical diagnosis of RSTS and *CREBBP/EP300* mutations negative carrying causal variants in three genes encoding members of epigenetic machinery but with other chromatin-modifying enzymatic mechanisms: *ASXL1* (two patients), *KMT2A* (seven patients) and *KMT2D* (one patient) [160,161]. *ASXL1* is a reader and is responsible for Bohring-Opitz syndrome (BOPS, OMIM #605039) and *KMT2A* et *KMT2D* (lysine methyltransferase 2A and 2D) are writers just like *CREBBP* et *EP300* and are responsible for Wiedemann-Steiner (WDSTS, OMIM #605130) and Kabuki (KS, OMIM #147920, #300867) syndromes respectively.

Woods et al. and Cucco et al. have identified *EP300* mutation in two patients with an initial diagnosis of Cornelia de Lange syndrome (CdLS, OMIM #122470, #300590, #610759, #614701, #300882, #608749) [162,163]. To date, six genes were implicated in the CdLS (*NIPBL*, *SMC1A*, *SMC3*, *RAD21*, *HDAC8*, and *BRD4*) and all of them encode proteins linked to the Cohesin complex [164,165] involved in chromatin organization and transcriptional regulation.

This great clinical overlap between the different chromatinopathies can be explained by the fact that genetic mutations within one of the components of this epigenetic machinery will modify the equilibrium between the opening and closing of the chromatin. The delicate balance between writers and erasers is thus disturbed and the modulations of the different histone marks on target genes within this interconnected network will lead to the different pathologies [166]. Many patients display typical clinical diagnoses such as an unidentified genetic cause or with a Variant of Unknown Significance (VUS). Others may have an atypical phenotype for which screening for genetic variants is difficult to guide.

The definition of epigenetic signatures opens new avenues for accurate diagnosis and clinical assessment in individuals affected by these disorders. In 2020, Aref-Eshghi et al. have evaluated DNA methylation episignatures in 42 Mendelian neurodevelopmental disorders [167]. These pathologies were selected a priori on the basis of the involvement of their associated genes in the epigenetic machinery and chromatin remodeling [168]. They identified 34 episignatures, including a specific RSTS signature associated with probes differentially methylated between patients and controls. This specific episignature helped to reclassify two patients with an uncertain diagnosis of RSTS. The first one had an RSTS phenotype but no pathogenic alteration in *CREBBP* or *EP300*. For the second, carrying a *de novo* VUS in *EP300* (c.92C > T; p.(Ile31Thr)), the RSTS diagnosis was discarded based on a methylation profile identical to the control group. These results need to be replicated on a larger cohort, especially to try to identify a subsignature to differentiate RSTS1-related *CREBBP* patients and RSTS2-related *EP300* patients.

## 6. Therapeutic Approaches

The reversible character of epigenetic marks brings real perspectives for the treatment of disorders of the epigenetic machinery, especially for intellectual disabilities.

As previously mentioned, an acetyl transferase deficiency can be compensated by the inhibition of HDAC proteins. The therapeutic approach by HDAC inhibitors (HDACi) appears, therefore, particularly interesting in RSTS (Figure 5). The HDACi belong to different pharmacological classes: hydroxamic acids (suberoylanilide hydroxamic acid (SAHA) and trichostatin A (TSA)), benzamides (tiapride or Tiapridal^®^), and carboxylic acids (sodium butyrate (NaB), phenylbutyrate and sodium valproate) [169,170]. Numerous studies have shown the involvement of these treatments in memory formation and the stabilization of certain hippocampal tasks such as the conditioned fear reflex or object recognition [142,146,171].

Regarding the RSTS, HDACi were tested in different mouse models: the haploinsufficient *Cbp*^+/−^ heterozygous mice [158], the *Cbp*^KIX/KIX^ homozygous KI mice, carrying a triple-point mutation in the KIX domain blocking the CBP-CREB interaction [155], the *Cbp* {HAT^−^} transgenic mice, in which the catalytic KAT domain has been rendered nonfunctional [172], and two models of cKO mice [152,156]. Intracerebroventricular injection of SAHA in *Cpb*^+/−^ mice both reversed H2B hypoacetylation in hippocampal area CA1 and restored cognitive functions assessed by contextual fear conditioning [158]. Injections of TSA restored object memory deficits in *Cbp* {HAT^−^} mice in association with increased H3 [172]. In addition, NaB injections improved object recognition memory in *Cbp*^KIX/KIX^ mice [173]. In contrast, in cKO mice, TSA administration failed to modulate learning and memory in mice in which CBP is completely inactivated in excitatory neurons of the postnatal forebrain [152] and NaB injections failed to restore the memory deficits induced by the focal deletion of CBP in the CA1 area of the hippocampus [156].

More recently, Babu et al. developed an *EP300* KO zebrafish model. This model causes severe defects reminiscent of RSTS such as smaller craniofacial structure, smaller eyes, shortened jaw, and reduced pectoral fins compared to controls. They performed a chemical screen and identified two lysine transferase inhibitors: HDACi III, an amide analog of TSA, and CHIC35, an inhibitor of SIRT1 lysine deacetylases. These compounds can partially correct craniofacial and pectoral fin cartilage defects in the RSTS model. These results open new therapeutic avenues for chromatinopathies [174].

In humans, data on the use of epidrugs are sparse. Lopez-Atalaya et al. showed that deficits in the levels of histone H2A and H2B acetylation in lymphoblastoid cell lines derived from two RSTS1-related *CREBBP* patients were rescued by treatment with TSA [175]. Furthermore, Parodi *et al.* very recently reviewed the overlap between chromatinopathies and fetal valproate syndrome, highlighting valproic acid as a promising therapeutic approach for RSTS [176]. However, these results should be confirmed in a therapeutic trial with a large cohort of patients as HDACi represent a therapeutic option for RSTS patients.

## 7. Conclusions and Perspectives

New generation sequencing techniques have improved the understanding of the genetic heterogeneity of the syndrome but also widened the phenotypic spectrum of RSTS by encompassing the broader field of chromatinopathies rendering phenotype/genotype correlations more complex (Figure 6). The emergence of multi-omics approaches, the integration of transcriptomic data coupled to DNA and histone modification profiles, and the development of patient-derived cellular models will likely contribute to a better definition of a specific epigenetic signature to the syndrome.

Indeed, less data are available on histone acetylation marks and their targets in RSTS. This is partly due to the fact that current mapping is focused on a limited number of marks on H3 and H4 and in particular H3K27ac and H3K9ac. However, recent work by Weinert et al. on the acetylome of RSTS mouse models showed that H2B was a major target of CBP/p300 [177]. These results corroborate the results of Lopez-Atalaya *et al.* on lymphoblastoid cell lines derived from RSTS patients, showing a global hypoacetylation of H2A and H2B compared to controls [175]. These individual signatures will serve as the basis for the implementation of new multi-omics diagnostic tools for RSTS but will also be applicable to other chromatinopathies and, in the longer term, for rational therapeutic design.

## Figures and Tables

**Figure 2 genes-12-00968-f002:**
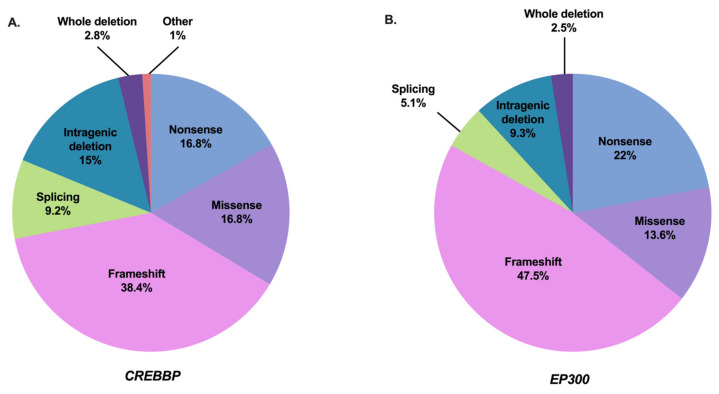
Mutation spectrum of *CREBBP* and *EP300* in RSTS individuals referenced in the literature and HGMDPro variant or LOVD databases [100,101]. (**A**) Repartition of all 500 pathogenic variants in *CREBBP* gene referenced as causing RSTS1 including 84 nonsense mutations, 192 frameshift mutations, 46 splicing mutations, 84 missense mutations, 75 intragenic deletions, 14 deletions including the entire *CREBBP* gene, and 5 other abnormalities (2 intragenic duplications and 3 complex rearrangements). (**B**) Repartition of all 118 pathogenic variants in *EP300* gene referenced as causing RSTS2 including 26 nonsense mutations, 56 frameshift mutations, 6 splicing mutations, 16 missense mutations, 11 intragenic deletions, and 3 deletions encompassing the entire *EP300* gene.

**Figure 3 genes-12-00968-f003:**
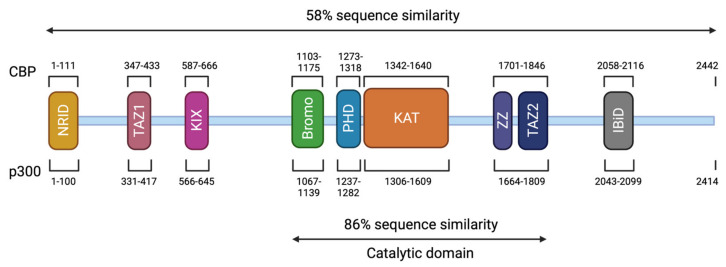
Structure of CBP and p300. The CBP protein is composed of 2442 amino acids (AA) and has a molecular weight of approximately 265 kDa. The p300 protein is composed of 2414 AA and has a molecular weight of approximately 265 kDa. These two proteins present 58% of sequence similarity within their domains. The different domains are represented and correspond to an N-terminal nuclear receptor interaction domain (NRID or RID), three cysteine-histidine rich regions (C/H1 and C/H3 domains contain zinc finger transcriptional adapters (TAZ1 and TAZ2), and C/H2 contains a homeodomain (PHD)), a KIX domain, a Bromodomain, a Lysine acetyltransferase domain (KAT) and an interferon-binding transactivation domain (IBiD). The position of the different domains is indicated relative to the position in the amino acid sequence. The catalytic domain has been highly conserved during evolution with 86% of similarity between CBP and p300 including the KAT domain and flanking regions.

**Figure 4 genes-12-00968-f004:**
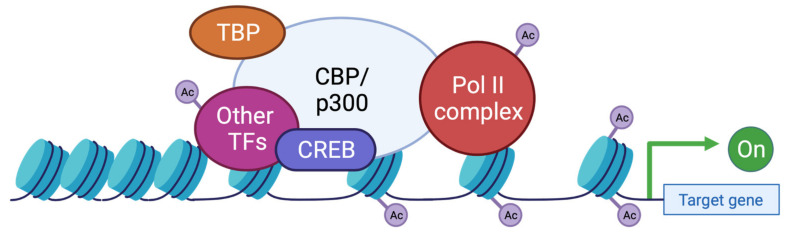
Functions of CBP/p300 as scaffold, bridge, and lysine acetyl transferase (KAT) activity. CBP and p300 act as transcriptional co-activators of target genes by different mechanisms: (1) Scaffolding function allowing the recruitment of transcription factors (TF) and in particular CREB; (2) Binding function by facilitating the physical and functional interactions of TFs; (3) KAT function by catalyzing the transfer of acetyl groups on lysine residues of both histone tails and non-histone proteins such as the RNApolIIcomplex and TFs. TBP: TATA binding protein; TF: transcription factor; Ac: acetyl group.

**Figure 5 genes-12-00968-f005:**
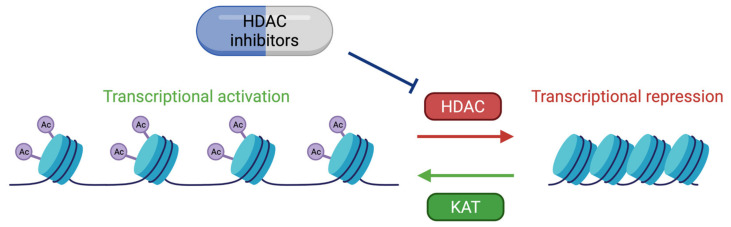
Therapeutic approach by HDAC inhibitors (HDACi). The action of the KAT proteins is antagonized by HDAC proteins allowing a dynamic regulation of the chromatin structure and the gene expression. Acetyl transferase deficiency can be compensated by HDAC inhibitors. HDACi tested in different RSTS mouse and in lymphoblastoid cell lines derived from RSTS patients rescue deficits in histone acetylation, particularly in H2A and H2B. HDACi represent a therapeutic option for improving intellectual efficiency in RSTS patients but have to be evaluated in clinical trials.

**Figure 6 genes-12-00968-f006:**
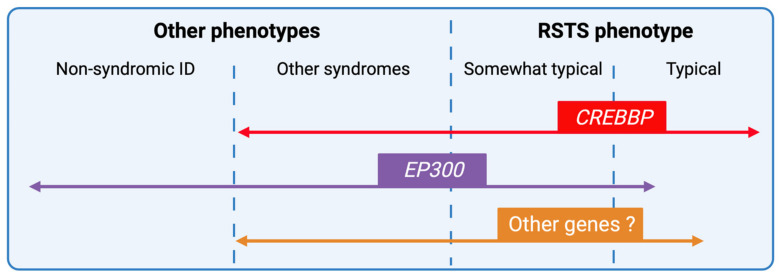
RSTS phenotypic overlap. Next-generation sequencing techniques have increased the number of genomic alterations identified and associated with RSTS phenotype and the phenotypic spectrum leading to complex phenotype/genotype correlations. *CREBBP*-mutated individuals present for the majority, a typical clinical diagnosis of RSTS, but the identification of missense mutations in exons 30 and 31 of the gene in patients with non-RSTS phenotype led to the definition of a new clinical entity: the Menke-Hennekam syndrome. *EP300*-mutated individuals display a wide clinical spectrum ranging from typical but milder RSTS phenotype to non-syndromic intellectual disability (ID), encompassing the phenotype of other chromatinopathies such as Cornelia de Lange syndrome. More recently, genes involved in others chromatinopathies (*ASXL1* for Bohring-Opitz syndrome, *KMT2A* for Wiedemann-Steiner syndrome, and *KMT2D* for Kabuki syndrome) have been identified in individuals with an RSTS phenotype but in the absence of *CREBBP/EP300* mutation. The definition of a specific epigenetic signature might reduce diagnostic deadlocks and open new therapeutic strategies.

**Table 1 genes-12-00968-t001:** Summary of the main phenotypic features in RSTS individuals carrying a *CREBBP* mutation compared to RSTS individuals with an *EP300* mutation reported in the literature. According to Fergelot et al., Yu et al., Pérez-Grijalba et al., Cross *et al.*, and Cohen et al. [17,21,37,99,126].

Phenotypic Features	*CREBBP* (*n* = 422)		*EP300* (*n* = 74)	
	Percentage	Number	Percentage	Number
Intrauterine growth retardation	25	55/220	43.1	25/58
Preeclampsia	3.4	2/59	25	16/64
Postnatal growth retardation	62.3	203/326	59.7	43/72
Microcephaly	52.7	129/245	82.4	61/74
Hypertrichosis	76.4	123/161	47.4	27/57
Facial dysmorphism				
Arched eyebrows	85.6	119/139	65.6	42/64
Long eyelashes	88.6	109/123	83.6	51/61
Downslanted palpebral fissures	81.1	258/318	51.6	33/64
Beaked nose	81.7	272/333	37.5	24/64
Columella below alae nasi	87.4	228/261	82.8	53/64
Highly arched palate	79.8	197/247	56.1	32/57
Micrognathia	64.2	149/232	40.6	26/64
Grimacing smile	94.9	112/118	36.8	21/57
Low-set ears	51.1	112/219	23.4	15/64
Broad thumbs/halluces	92.3	373/404	59.5	44/74
Angulated thumbs	56.4	184/326	4.8	3/63
Intellectual disability	82.2	287/349	84.9	62/73
Severe	35.9	33/92	7.3	3/41
Moderate	47.8	44/92	26.8	11/41
Mild	14.1	13/92	65.9	27/41
Autism/Behavioral problems	49.4	78/158	21.3	13/61
Cardiovascular anomalies	34.5	99/287	29	20/69
Urinary tract anomalies	37.4	61/163	26.3	15/57

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
