# Peer review of "Rubinstein-Taybi Syndrome: A Model of Epigenetic Disorder"

_genes, 2021, doi:10.3390/genes12070968_

Round 1
Reviewer 1 Report
The manuscript by Van Gils and colleagues reviews clinical and molecular aspects of Rubinstein-Taybi syndrome, a model of chromatinopathy.
The Authors describe all clinical signs showed by RSTS patients, ranging from the more frequent to the ones occurring in few cases. They report pathogenic variants from two databases (HGMDPro variant and LOVD) and correlate genotype with phenotype of patients. In addition, they discuss the role of epigenetic actors in neurodevelopment and cognitive function, hinting at other related chromatinopathies. Authors conclude presenting publications on therapeutic approaches for RSTS and highlighting the necessity of multi-omics strategies for this syndrome and other chromatinopathies.
In order to improve the manuscript, the following changes are suggested:
- paragraph on related chromatinopathies should be completed adding considerations about the phenotype of the cited syndromes;
- the Authors described GADVES as a chromatinopathy (line 488). To date, YY1 is not considered a member of the epigenetic machinery. If Authors support this thesis, please argue it;
- the part regarding the therapeutic approaches needs to be updated (e.g. Babu et al 2018, Di Fede et al 2021);
- clinical description (paragraph 2) could be divided into subsections to help the readers;
- figures can be modernized (Figure 4 can be removed and Figure 5 should be modified as it is very similar to the one in Chuang et al 2009).
Additional minor revisions:
-Line 46 Please add a reference for “Chromatinopathies”.
-Line 57 Please write the acronym HTA in full.
-Line 65 Diagnosis of intellectual disability can be made in childhood.
-Line 74 Capillary hemangioma appears as one of the main clinical signs in the text. Please rephrase as a possible association.
-Line 99 Spina bifida should be classified as neurodevelopmental defects (i.e. neural tube defects) rather than a skeletal anomaly.
-Lines 101-118 An interesting and peculiar neurological aspect of RSTS patients that could be mentioned is the fluid reasoning higher than IQ (Ajmone et al 2018).
-Line 164 Please correct “carries” with “caries”.
-Line 172 A dot is missing after “period”.
-Line 183 Incidence of glabellar hemangioma seems overestimated, please add a more recent reference.
-Line 185 Please correct “Saetinni” with “Saettini”.
-Lines 202-203 “Expressivity” is not appropriate, please rephrase.
-Line 230 “Responsible” is probably a typo.
-Line 239 Authors in the text always wrote KAT domain except at this point, please change accordingly.
-Line 254 Please correct “is” with “are”.
-Lines 256-268 Since Authors mentioned KAT3 family, please indicate the other name of CBP and p300 accordingly.
-Lines 354-357 Please add EP300 referring to Menke-Hennekam syndrome.
-Line 357 Please rephrase “Patients present a distinct RSTS phenotype”, as Authors at this point are discussing about a different syndrome, which is not RSTS.
-Lines 381-390 In mouse model described in Oike et al 1999 are present also cardiac anomalies, please add.
-Lines 458-463 Please rephrase as the sentence is too long and hard to read.
-Lines 472-473 The sentence is not clear, please rephrase.
-Line 484 The name of the gene is SRCAP, please correct.
-Line 490 Please correct “Fedi” with “Di Fede”.
-Line 500 The name of the gene is NIPBL, please correct.
-Line 505 Please remove “fiber”.
-Lines 508-510 The sentence is difficult to read, please rephrase.
-Line 514 and 520 Please write “a priori” and “de novo” in italics.
-Line 528 It is better to write HDACi (without s) even when it is plural, please change the text accordingly.
-Line 562 Please remove “improving intellectual efficiency in”.
-Line 576 Please correct “Loper” with “Lopez”.
-Please check paragraph 5.4 as all genes names should be written in italics and change them accordingly.
-Please add in the figure legend of Figure 1 respective references and in Figure 1A the age of the patients.
-Please add in the figure legend of Figure 2A what “Others” stands for.
-In Table 1 should be added other peculiar signs such as ptosis, strabismus, other skeletal anomalies aside from broad thumbs/halluces (e.g. delayed bone age), hypotonia and, mostly, IUGR and pre-eclampsia as Authors pointed out these features in the context of genotype-phenotype correlation.
-Please make nomenclature in Supplementary Tables S1-S3 homogeneous. Authors should use HGVS recommendations (e.g. “*” instead of “Term”).
Author Response
Response to Reviewer 1
Reply to Reviewer
We thank the reviewer for his/her careful evaluation of the manuscript and helpful suggestions for improving the text. We hope that our point-by-point responses will satisfy his/her requests and that the manuscript will be suitable for publication.
Comments to the Authors:
The manuscript by Van Gils and colleagues reviews clinical and molecular aspects of Rubinstein-Taybi syndrome, a model of chromatinopathy.
The Authors describe all clinical signs showed by RSTS patients, ranging from the more frequent to the ones occurring in few cases. They report pathogenic variants from two databases (HGMDPro variant and LOVD) and correlate genotype with phenotype of patients. In addition, they discuss the role of epigenetic actors in neurodevelopment and cognitive function, hinting at other related chromatinopathies. Authors conclude presenting publications on therapeutic approaches for RSTS and highlighting the necessity of multi-omics strategies for this syndrome and other chromatinopathies.
In order to improve the manuscript, the following changes are suggested:
- paragraph on related chromatinopathies should be completed adding considerations about the phenotype of the cited syndromes.
Response: We thank the reviewer for this comment. The objective of this paragraph was mainly to link the common physiopathological mechanisms of these different pathologies. Phenotypic overlaps are indeed present, but they are well detailed in the cited publications (Larizza et al., Negri et al., Fedi et al., Woods et al. and Cucco et al. Due to space limitation, we have chosen not to add the phenotypes of each syndrome that would make the paragraph too heavy and to remain focused on the description of RSTS. Readers are invited to refer to other reviews mentioned in the text.
- the Authors described GADVES as a chromatinopathy (line 488). To date, YY1 is not considered a member of the epigenetic machinery. If Authors support this thesis, please argue it;
Response: We agree that YY1 is not per se a chromatin factor but a transcription factor. This gene was indicated as a key partner in translating chromatin structure to gene expression (Larizza & Finelli, 2019). The sentence has been modified: “Gabrielle-De Vries syndrome (GADEVS, OMIM #617557) (mutations of YY1 gene) has been also associated. GADEVS is not per se a chromatinopathie but YY1 act indirectly on epigenetic machinery as a key partner encoding transcriptional regulator interacting with CBP/p300)[162].” (Lines 516-519).
- the part regarding the therapeutic approaches needs to be updated (e.g. Babu et al 2018, Di Fede et al 2021);
Response: A short paragraph has been added showing the therapeutic perspectives provided by the zebrafish model of Babu et al. (Lines 590-596). Furthermore, we mentioned the very recent review by Parodi et al. On the overlap between chromatinopathies and fetal valproate syndrome that highlights Valproic acid as a promising therapeutic approach for RSTS (Lines 600-602).
- clinical description (paragraph 2) could be divided into subsections to help the readers.
Response: clinical description has been divided into 6 subsections as suggested
- figures can be modernized (Figure 4 can be removed and Figure 5 should be modified as it is very similar to the one in Chuang et al 2009).
Response: Regarding Figure 4, the goal was to illustrate in a simplistic manner the main involvement of CBP/p300 in chromatin organization, in particular for readers who might not be used to chromatin-related mechanisms. So we preferred to keep it. Figure 5 has been modified.
Additional minor revisions:
-Line 46 Please add a reference for “Chromatinopathies”.
Response: We agree with the reviewer. The reference has been added: “The Rubinstein-Taybi syndrome is a developmental disorder whose physiopathology are based primarily on an epigenetic mechanism, belonging thereby to the group of “Chromatinopathies” defined as Mendelian disorders of the epigenetic machinery, as reviewed in [15] (Fahrner et al. 2019). (Lines 44-47).
-Line 57 Please write the acronym HTA in full.
Response: We agree with the reviewer. We have replaced “HTA” by “hypertension”.
-Line 65 Diagnosis of intellectual disability can be made in childhood.
Response: The clarification is made at the end of the paragraph: “The diagnosis is most often made at birth or in early childhood by observing the classic association of post-natal growth retardation, characteristic facial dysmorphism, broad thumbs and halluces and intellectual disability.” (Lines 66-68).
-Line 74 Capillary hemangioma appears as one of the main clinical signs in the text. Please rephrase as a possible association.
Response: We thank the reviewer for this comment. The sentence has been modified: “A capillary hemangioma is also often described.” (Line 81).
-Line 99 Spina bifida should be classified as neurodevelopmental defects (i.e. neural tube defects) rather than a skeletal anomaly.
Response: We thank the reviewer for this observation. Spina bifida has been removed from the skeletal anomalies paragraph and added to the neurological disorders paragraph.
-Lines 101-118 An interesting and peculiar neurological aspect of RSTS patients that could be mentioned is the fluid reasoning higher than IQ (Ajmone et al 2018).
Response: We thank the reviewer for this comment. A sentence has been added in this sense: “An interesting and peculiar neurological aspect of RSTS patients is that the fluid reasoning is higher than IQ showing a more flexible cognitive ability in these indiviuals.” (Lines 115-117).
-Line 164 Please correct “carries” with “caries”.
Response: The typo has been corrected.
-Line 172 A dot is missing after “period”.
Response: The missing dot has been added.
-Line 183 Incidence of glabellar hemangioma seems overestimated, please add a more recent reference.
Response: To the best of our knowledge, no recent publication with a large molecularly confirmed cohort has reported the incidence of glabellar hemangioma. For this reason, we have removed the precision on the incidence of this feature (Line 193).
-Line 185 Please correct “Saetinni” with “Saettini”.
Response: The typo has been corrected.
-Lines 202-203 “Expressivity” is not appropriate, please rephrase.
Response: We agree with the reviewer. We have replaced “Expressivity” by “clinical heterogeneity”.
-Line 230 “Responsible” is probably a typo.
Response: The sentence has been modified: “Abnormalities in the EP300 gene are responsible of about 8-11% of cases.” (Line 258).
-Line 239 Authors in the text always wrote KAT domain except at this point, please change accordingly.
Response: This change has been made.
-Line 254 Please correct “is” with “are”.
Response: “is composed of” has been replaced by “contain”: “CREBBP and EP300 contain 31 exons.” (Line 282).
-Lines 256-268 Since Authors mentioned KAT3 family, please indicate the other name of CBP and p300 accordingly.
Response: KAT3A for CBP and KAT3B for p300 have been added (Line 283).
-Lines 354-357 Please add EP300 referring to Menke-Hennekam syndrome.
Response: We thank the reviewer for this observation. EP300 has been added to the definition of Menke-Hennekam syndrome (Line 385).
-Line 357 Please rephrase “Patients present a distinct RSTS phenotype”, as Authors at this point are discussing about a different syndrome, which is not RSTS.
Response: The sentence has been corrected for more clarity: “Patients present a different syndrome which is not RSTS.” (Line 386).
-Lines 381-390 In mouse model described in Oike et al 1999 are present also cardiac anomalies, please add.
Response: We thank the reviewer for this observation. The precision has been added: “In addition to skeletal abnormalities, heterozygous mutants also exhibit growth retardation[139–142], increased insulin sensitivity[143], cardiac malformations[139] and disorders of hematopoiesis with a tendency to develop hematologic malignancies[142].” (Lines 415-417).
-Lines 458-463 Please rephrase as the sentence is too long and hard to read.
Response: The sentence was corrected and clarified: “Transcriptomic analysis on these RSTS i-neurons revealed two levels of dysregulation during neuronal differentiation. The first concerns “active” gene modulation associated with aberrant upregulation of genes involved in neural migration and axonal and dendritic targeting and downregulation of RNA and DNA metabolic genes. The second relates to a “passive” gene modulation by de-regulation of some genes involved in synaptic integration compared to controls[136].” (Lines 486-491).
-Lines 472-473 The sentence is not clear, please rephrase.
Response: The sentence was corrected and clarified: “Alterations of These genes encode “writers” (responsible for establishing a mark on either DNA or histone tail), “erasers” (removal or modification of written marks), “readers” (mediate the interaction of the mark with a protein complex to enhance transcription) or “remodelers” (alter the accessibility of chromatin through ATP-dependent sliding of nucleosomes along DNA).” (Lines 500-505).
-Line 484 The name of the gene is SRCAP, please correct.
Response: The typo has been corrected.
-Line 490 Please correct “Fedi” with “Di Fede”.
Response: The typo has been corrected.
-Line 500 The name of the gene is NIPBL, please correct.
Response: The typo has been corrected.
-Line 505 Please remove “fiber”.
Response: The word has been removed.
-Lines 508-510 The sentence is difficult to read, please rephrase.
Response: The sentence was corrected and simplified: “Many patients display typical clinical diagnosis with unidentified genetic cause or with Variant of Unknown Significance (VUS). Others may have an atypical phenotype for which screening for genetic variants is difficult to guide.” (Lines 539-541).
-Line 514 and 520 Please write “a priori” and “de novo” in italics.
Response: We thank the reviewer for this observation. These changes have been made.
-Line 528 It is better to write HDACi (without s) even when it is plural, please change the text accordingly.
Response: These changes have been made.
-Line 562 Please remove “improving intellectual efficiency in”.
Response: The sentence has been removed.
-Line 576 Please correct “Loper” with “Lopez”.
Response: The typo has been corrected.
-Please check paragraph 5.4 as all genes names should be written in italics and change them accordingly.
Response: We thank the reviewer for this observation. All gene names have been rewritten in italics
-Please add in the figure legend of Figure 1 respective references and in Figure 1A the age of the patients.
Response: In Figure 1A the age of the patients has been added. The patient on the left was previously reported by Lacombe et al. The patients in the middle and on the right have not been reported in any publication to date This clarification has been added.
-Please add in the figure legend of Figure 2A what “Others” stands for.
Response: The precision has been added in the Figure 2A legend.
-In Table 1 should be added other peculiar signs such as ptosis, strabismus, other skeletal anomalies aside from broad thumbs/halluces (e.g. delayed bone age), hypotonia and, mostly, IUGR and pre-eclampsia as Authors pointed out these features in the context of genotype-phenotype correlation.
Response: Table 1 summarizes the clinical signs of RSTS1 vs RSTS2 patients described in the cohorts of Fergelot et al., Yu et al., Pérez-Grijalba et al., Cross et al. and Cohen et al.
These publications were chosen because they include the most recent data on large cohorts with only molecularly identified patients. Information about IUGR and preeclampsia was added although information was missing for Pérez-Grijalba et al. and Cross et al.
It does not seem relevant to add the other signs because only a few data are available to date. Indeed, delayed bone age and hypotonia are not reported in any of the 5 publications cited. The ptotis is specified only by Yu et al (8/14 of CREBBPpatients). Finally, ocular abnormalities (without more precision) are reported for CREBBP patients, but the information is missing for EP300 patients.
-Please make nomenclature in Supplementary Tables S1-S3 homogeneous. Authors should use HGVS recommendations (e.g. “*” instead of “Term”).
Response: According to the reviewer’s comment, we have modified the nomenclature according to HGVS recommendations. The nomenclature has been standardized between tables S1, S2 and S3. “Aminochange” has been changed to “Protein”.

Reviewer 2 Report
A very thorough review on all the main issues of the syndrome. In my opinion, the paper is suitable for publication, I have only some minor comments:
Paragraph 2:
please change “minor facial criteria” with “minor facial features”
please change “a few signs of antenatal signs” with “a few antenatal signs”
In line 104, “more moderate intellectual disability” could be confusing: I think “milder intellectual disability” sound better
“Classically, intrauterine growth and birth measurements (weight, height and occipital frontal circumference (OFC)) appear classically around the 50th percentile.” Please erase one of the repeated words “classically”
“A delay in bone age is classically associated (74%)”, you can replace “classically” with “often” or similar.
Line 130 you can write RSTS instead of Rubinstein Taybi Syndrome
Line 164, there is a typo (“carries”)
Line 172: “colobomas, iris, retina or optic nerve” please change in “colobomas of…”
Line 176: “will develop”; I think you could change in “cryptorchidism is described in…”. The meaning of the verb to develop doesn’t fit well with this medical issue.
Line 185, a typo “Saetinni”, replace with “Saettini”
Paragraph 3:
Lines 219-220 and 233-234, there are some calc mistakes
Line 230: please change in “Abnormalities in the EP300 gene are responsible of about 8-11% of cases”
Line 239-242: “Only three patients with RSTS have been reported in the literature with a missense mutation in EP300; however, each of these mutations was inherited from a healthy parent, making the pathogenic involvement of these variants difficult” I guess you mean out of KAT domain, is it correct?
Line 254: “CREBBP and EP300…” please change “is” in “are”
Paragraph 5:
Line 472-473: please erase “Alterations of …”
Author Response
Response to Reviewer 2
Reply to Reviewer
We thank the reviewer for his/her careful evaluation of our manuscript and helpful suggestions for improving the text. We hope that our point-by-point responses will be satisfactory and that the revised manuscript will be acceptable for publication.
Comments to the Authors:
A very thorough review on all the main issues of the syndrome. In my opinion, the paper is suitable for publication, I have only some minor comments:
Paragraph 2:
please change “minor facial criteria” with “minor facial features”
Response: This change has been made.
please change “a few signs of antenatal signs” with “a few antenatal signs”
Response: This change has been made.
In line 104, “more moderate intellectual disability” could be confusing: I think “milder intellectual disability” sound better
Response: We agree with the reviewer. This change has been made.
“Classically, intrauterine growth and birth measurements (weight, height and occipital frontal circumference (OFC)) appear classically around the 50th percentile.” Please erase one of the repeated words “classically”
Response: The first “classically” has been erased.
“A delay in bone age is classically associated (74%)”, you can replace “classically” with “often” or similar.
Response: This change has been made.
Line 130 you can write RSTS instead of Rubinstein Taybi Syndrome
Response: This change has been made.
Line 164, there is a typo (“carries”)
Response: The typo has been corrected.
Line 172: “colobomas, iris, retina or optic nerve” please change in “colobomas of…”
Response: The missing word has been added
Line 176: “will develop”; I think you could change in “cryptorchidism is described in…”. The meaning of the verb to develop doesn’t fit well with this medical issue.
Response: We thank the reviewer for this comment. This change has been made.
Line 185, a typo “Saetinni”, replace with “Saettini”
Response: The typo has been corrected.
Paragraph 3:
Lines 219-220 and 233-234, there are some calc mistakes
Response: We thank the reviewer for this comment. We agree with the reviewer that on the HGDD and LOVD databases, more variants are referenced for CREBBP and EP300.
However, many variants are associated with other phenotypes than RSTS like Menke-Hennekam syndrome for example. Moreover, different classes of pathogenicity of variants are reported. However, as stated in the manuscript, we want to reference only the variants that are clearly pathogenic and associated with the RSTS1 phenotype for CREBBP or RSTS2 for EP300: “To date, 500 pathogenic variants in this gene have been referenced as causing RSTS1 (55 of which are unpublished)”; “To date, 118 pathogenic variants in this gene have been referenced as causing RSTS2 (8 of which are unpublished)”.
We therefore filtered the databases to obtain 500 pathogenic variants associated with RSTS1 for CREBBP (545 in HGMD and 45 unpublished in LOVD) and 118 pathogenic variants associated with RSTS2 for EP300 (110 in HGMD and 18 unpublished in LOVD).
Line 230: please change in “Abnormalities in the EP300 gene are responsible of about 8-11% of cases”
Response: This change has been made.
Line 239-242: “Only three patients with RSTS have been reported in the literature with a missense mutation in EP300; however, each of these mutations was inherited from a healthy parent, making the pathogenic involvement of these variants difficult” I guess you mean out of KAT domain, is it correct?
Response: We agree with the reviewer for this observation. The clarification has been made: “Only three patients with RSTS have been reported in the literature with a missense mutation in EP300 out of KAT domain.” (Lines 267-269).
Line 254: “CREBBP and EP300…” please change “is” in “are”
Response: “is composed of” has been replaced by “contain”: “CREBBP and EP300 contain 31 exons.” (Line 282).
Paragraph 5:
Line 472-473: please erase “Alterations of …”
Response: The words have been erased.

Reviewer 3 Report
Van Gils et al. reported a review about RSTS syndrome as a model of epigenetic disorder.
In my opinion, the manuscript is well structured, easy and clear to understand, and gives a global vision of the different aspects of RSTS syndrome as episyndrome.
However, I would like to suggest that:
- In table S1 in amino acid change column, follow the same nomenclature as the rest of the tables. Example: p.(Arg14Gly)
Author Response
Response to Reviewer 3
Reply to Reviewer
We thank the reviewer for his/her careful and positive evaluation of our manuscript and helpful suggestions for improving the text. We hope that our point-by-point responses will be satisfactory that the improved manuscript will be suitable for publication.
Comments to the Authors:
Van Gils et al. reported a review about RSTS syndrome as a model of epigenetic disorder.
In my opinion, the manuscript is well structured, easy and clear to understand, and gives a global vision of the different aspects of RSTS syndrome as episyndrome.
However, I would like to suggest that:
In table S1 in amino acid change column, follow the same nomenclature as the rest of the tables. Example: p.(Arg14Gly)
Response: The nomenclature has been standardized between tables S1, S2 and S3. “Aminochange” has been changed to “Protein”.
